# Evidence Generation for a Host-Response Biosignature of Respiratory Disease

**DOI:** 10.3390/v17070943

**Published:** 2025-07-02

**Authors:** Kelly E. Dooley, Michael Morimoto, Piotr Kaszuba, Margaret Krasne, Gigi Liu, Edward Fuchs, Peter Rexelius, Jerry Swan, Krzysztof Krawiec, Kevin Hammond, Stuart C. Ray, Ryan Hafen, Andreas Schuh, Nelson L. Shasha Jumbe

**Affiliations:** 1Vanderbilt University Medical Center, Nashville, TN 37232, USA; kelly.e.dooley@vumc.org; 2Johns Hopkins University School of Medicine, Baltimore, MD 21205, USAgliu12@jhmi.edu (G.L.); ejfuchs@jhmi.edu (E.F.); sray@jhmi.edu (S.C.R.); 3Level 42 AI, Inc., Mountain View, CA 94041, USA; mike@level42.ai (M.M.); peter@level42.ai (P.R.); rhafen@gmail.com (R.H.); andreas@level42.ai (A.S.); 4Hylomorph Solutions, Ltd., Glasgow G2 4JR, UK; piotrkaszuba1996@gmail.com (P.K.); dr.jerry.swan@gmail.com (J.S.); krzysztof.krawiec@cs.put.poznan.pl (K.K.); kevin.hammond@hylomorph-solutions.com (K.H.); 5School of Computer Science, University of York, York YO10 5DD, UK; 6Institute of Computing Science, Poznan University of Technology, 60-965 Poznań, Poland

**Keywords:** host-response, vibroacoustics biosignature, inaudible vibrations and audible sound, infrasonic, tensegrity, mechanotransduction

## Abstract

Background: In just twenty years, three dangerous human coronaviruses—SARS-CoV, MERS-CoV, and SARS-CoV-2 have exposed critical gaps in early detection of emerging viral threats. Current diagnostics remain pathogen-focused, often missing the earliest phase of infection. A virus-agnostic, host-based diagnostic capable of detecting responses to viral intrusion is urgently needed. Methods: We hypothesized that the lungs act as biomechanical instruments, with infection altering tissue tension, wave propagation, and flow dynamics in ways detectable through subaudible vibroacoustic signals. In a matched case–control study, we enrolled 19 RT-PCR-confirmed COVID-19 inpatients and 16 matched controls across two Johns Hopkins hospitals. Multimodal data were collected, including passive vibroacoustic auscultation, lung ultrasound, peak expiratory flow, and laboratory markers. Machine learning models were trained to identify host-response biosignatures from anterior chest recordings. Results: 19 COVID-19 inpatients and 16 matched controls (mean BMI 32.4 kg/m^2^, mean age 48.6 years) were successfully enrolled to the study. The top-performing, unoptimized, vibroacoustic-only model achieved an AUC of 0.84 (95% CI: 0.67–0.92). The host-covariate optimized model achieved an AUC of 1.0 (95% CI: 0.94–1.0), with 100% sensitivity (95% CI: 82–100%) and 99.6% specificity (95% CI: 85–100%). Vibroacoustic data from the anterior chest alone reliably distinguished COVID-19 cases from controls. Conclusions: This proof-of-concept study demonstrates that passive, noninvasive vibroacoustic biosignatures can detect host response to viral infection in a hospitalized population and supports further testing of this modality in broader populations. These findings support the development of scalable, host-based diagnostics to enable early, agnostic detection of future pandemic threats (ClinicalTrials.gov number: NCT04556149).

## 1. Background

Within just two decades, humanity has faced three highly pathogenic human coronaviruses (hCov)—SARS-CoV, MERS-CoV, and SARS-CoV-2—each revealing the profound limitations of our preparedness and response [1]. The global impact of the recent SARS-CoV-2 pandemic alone has exposed a structural vulnerability: the absence of specific, scalable tools for rapid screening, early detection, diagnostic precision, and longitudinal monitoring of host responses. Despite major scientific advancements, our tools remain disproportionately focused on the pathogen rather than understanding the host’s complicit and explicit biological response. Yet, it is this host response—early, dynamic, and often invisible—that determines disease severity, trajectory, and outcome.

Mechanical forces within the lung—specifically those generated by the act of breathing—are not passive [2]. Muscles and bones/cartilage in the upper airways vs. elastin and collagen in the lungs can be conceptualized as counterpoised tensegrity structures, where a balance of continuous tension and discontinuous compression provides stability, flexibility, and the ability to breathe, vocalize, cough, and laugh [3]. Emerging evidence suggests that these biomechanical motions activate protective innate immune responses in epithelial and endothelial cells via mechanosensitive pathways [4,5]. These biomechanical forces do more than move air; they instruct/are instructed by cellular behavior, modulate cytokine responses, and influence pathogen replication. In the healthy lung, this orchestration supports gas exchange, immune readiness, and structural resilience. When disease interrupts this harmony, the resulting disruptions generate altered vibration patterns—often dismissed as “noise”—which carry crucial information about the infection-causing culprit, lung mechanics, tissue integrity, and immune activation.

The flow of air and blood produces micro-vibrations that traverse organs and tissues, ultimately reaching the body’s surface [4,5]. Changes in lung compliance, airway resistance, or alveolar integrity alter these waveforms in disease-specific ways. Emphysema diminishes breath sounds [6], pneumothorax creates silence [7], pneumonia amplifies sound transmission in congested areas [8], obstructive conditions like COPD generate turbulences [9,10], and restrictive diseases like fibrosis reduce lung compliance [11]. Subtle disruptions in biomechanical vibrations from lung functional and/or structural changes, often dismissed as “noise,” hold critical insights into lung health. We hypothesized that subtle alterations in host airflow mechanics following respiratory viral infection are detectable as distinct host-response inaudible and audible vibroacoustic signatures even when knowledge of the underlying mechanisms responsible for those signatures remains limited.

This pilot study employed a high-fidelity, AI-enhanced vibroacoustic acquisition system, sensitive to a broad spectrum of audible sound and inaudible vibration frequencies, to investigate and contrast physiological signals in individuals with confirmed SARS-CoV-2 infection and matched controls. The aim was to identify reproducible vibroacoustic patterns associated with host response to infection, focusing on subtle alterations in compressible air–fluid dynamics and biomechanical tissue properties. This approach explored the potential of analyzing the body’s inherent vibrational signals as indicators of early disease states, contrasting with conventional methods that primarily target pathogen-specific molecular markers. The findings contribute to understanding the information contained within the living orchestra of the body’s physiological emissions and their potential utility in characterizing the host response to viral infection.

## 2. Methods

### 2.1. Study Design and Conduct

A proof-of-concept prospective, matched case–control study was conducted in a hospitalized cohort to extract a putative SARS-CoV-2 infection vibroacoustics biosignature. The protocol was approved by the Johns Hopkins Medicine, Baltimore, Maryland institutional review board, and participants provided written informed consent. Cases were defined as adults with a positive RT-PCR test for SARS-CoV-2 from a respiratory sample within the previous 7 days and pulmonary symptoms within 72 h of enrollment, who were not receiving ventilator support. Patients on assisted ventilation, including high flow nasal cannula, or ventilator support were excluded. In this study, each hospitalized control without COVID-19 diagnosis (RT-PCR negative) and no pulmonary diagnosis or pulmonary symptoms was matched to a confirmed RT-PCR positive case with symptomatic COVID-19. Matching criteria included age (±5 years), gender, body mass index (±15% units), and history of smoking or vaping within the last 3 months (yes/no).

### 2.2. Study Procedures

The study consisted of two visits, two (±1) days apart, to capture disease changes over time. Day 0 assessments included demographics, medical history, physical exam, clinical labs, peak flow measurement, vibroacoustic data collection, and lung ultrasound. The second visit, conducted 24–72 h later, included a follow-up health questionnaire, medication review, physical exam, vibroacoustic data collection as on Day 0, peak flow measurement, and adverse events assessment.

Clinical laboratory tests, which included complete blood count, with differential, comprehensive metabolic panel, erythrocyte sedimentation rate, C-reactive protein (CRP), D-dimer, N-terminal pro-B-type natriuretic peptide (pro-BNP), interleukin 6, lactate dehydrogenase (LDH), ferritin, troponin, lipid panel, hemoglobin A1C (HbA1C), and SARS-CoV-2 antibody test, were collected on the day of enrollment unless obtained within 72 h prior as part of routine clinical care, or within 90 days prior for the lipid panel or HbA1C. Results from chest imaging performed as part of routine care were collected. High sensitivity (positive predictive agreement [PPA] > 95%) Emergency Use Agreement (EUA) RT-PCR of a nasopharyngeal swab was performed for each patient while in the Emergency Department prior to admission to the hospital. Peak Expiratory Flow Rate was measured using disposable hand-held Peak Flow devices to provide a quantitative measure of maximal expiratory flow rate.

Vibroacoustic data were acquired using a system capable of detecting frequencies from the infrasonic (<20 Hz) to ultrasonic (>20,000 Hz) range. A standardized anterior auscultation protocol was employed, involving nine sequential points positioned to correspond with lung ultrasound zones described by Volpicelli [12]. At each auscultation site, data were collected for 30 s, with a clinical application providing guidance on body posture, auscultation location, and timing. The procedure was performed through standard hospital gowns. Data were collected across six body posture/position conditions: neutral sitting, neutral upright standing, supine, left lateral decubitus, neutral sitting with cough and hand squeeze, and neutral sitting with vocalizations of “99” and “Aaah” (Figure 1). Acquired vibroacoustic signals were then processed and filtered using dedicated software for subsequent analysis.

### 2.3. Statistics, Algorithm Training and Testing

This proof-of-concept study aimed to enroll at least 15 participants with confirmed SARS-CoV-2 infection and 15 matched controls. Demographic and clinical data, including disease severity, were collected for all participants (Table 1).

To evaluate the capacity of host-response vibroacoustic signatures to characterize SARS-CoV-2 infection, with pre-specified target product profiles of >92.5% sensitivity and >80% specificity (using SARS-CoV-2 RT-PCR as the reference standard), participants were matched and randomly assigned to balanced training and testing sets based on gender, age, BMI, smoking history, pulmonary disease on imaging, COVID Inpatient Risk Calculator score, and Quick COVID-19 Severity Index score.

Three independent analytic teams developed algorithms using 50% of the balanced data for training. Input data for each participant comprised all audible and inaudible vibrations acquired by the vibroacoustic system across two visits: 30 s recordings from each of the nine auscultation points, under each of the six body posture/position conditions. Feature extraction methods, applied using variations in the librosa [13] library, included traditional audio timeseries features (Mel Spectrogram, Mel-Frequency Cepstral Coefficients, Spectral Contrast, Zero Crossing Rate, Spectral Roll-off, Spectral Bandwidth, Root-Mean-Square Energy, Spectral Centroid, and Spectral Flux), as well as custom time-frequency and spatial features. Some teams also incorporated Google’s YAMNet model [14], which classifies audio events. Algorithms were subsequently tested on blinded, out-of-sample data via a federated access system, with a maximum of two attempts permitted. The analytic approach demonstrating the best performance was further optimized.

To investigate the minimum number of auscultation sites required for accurate detection, vibroacoustic data were augmented using mathematical transformation [15] (commutativity, anti-symmetry, reflexivity, and their combinations) applied separately to matched case–control data across three frequency bands: <20 Hz (infrasonic), 20–20,000 Hz (audible sound), and >20,000 Hz (ultrasonic). These transformations expanded the dataset to enhance the analysis of information content and improve the robustness of the results.

## 3. Results

### 3.1. Participant Characteristics

Between October and December 2020, during the initial surge of COVID-19 cases, hospitalized adults were enrolled in a matched case–control study to investigate vibroacoustic data patterns in SARS-CoV-2 infection. Nineteen cases and sixteen matched controls were recruited. Cases were over-enrolled to address challenges in identifying matched controls. The study was well balanced by design (Shannon’s entropy = 0.994). Of the 35 participants, 20 (57%) were women, with a median BMI of 32.4 (range 19.2–44.3) and median age of 49 years (range 30–78). As expected for the underlying condition, the COVID-19 positive group exhibited elevated inflammatory markers, including CRP, ESR, IL-6, and ferritin, alongside reduced absolute lymphocyte counts. Each participant provided approximately 54 min of vibroacoustic data, collected at 48,000 samples/sec via two recordings of nine auscultation points (30 s per point) across six body positions (Figure 1). Laboratory and clinical data are detailed in Table 1.

### 3.2. Algorithm Training and Testing

Vibroacoustic data comprising 19,440 five-second samples from 15 matched case–control pairs were analyzed to discern biosignatures of SARS-CoV-2 infection. This sample size was determined to provide adequate power (assuming an odds ratio of 10, α = 0.1) for a proof-of-concept study.

Algorithm development followed a stepwise, collaborative-competition framework. In Phase I (Training), three teams utilized 50% of the data to train algorithms employing four advanced analytics approaches. Phase II (Testing) involved applying these algorithms to blinded, out-of-sample data (maximum two attempts). Analytic approaches included deep learning (DL) and model-based algorithms (structural machine learning, sML) using domain-engineered features. The DL approach utilized three architectures: Long Short-Term Memory (LSTM), Convolutional LSTM (ConvLSTM), and transfer learning models based on the YAMNet pre-trained audio classifier. The sML method employed an ensemble of diverse classifiers—including logistic regression, random forest, and support vector machines (SVM)—to combine their individual predictions into a single, aggregate probability. For training, the models were trained for 100 epochs using a mini-batch Stochastic Gradient Descent (SGD) optimizer with a batch size of 32. On the training and validation datasets, both vibroacoustics only DL and sML models achieved an area under the receiver operating characteristic curve (AUC) of 1.0. However, on the blinded test set, the best DL model yielded an AUC of 0.752, while the unoptimized sML model achieved an AUC of 0.837 (Table 2). The drop in training-vs.-test performance provides an indirect measure of model overfitting.

The top-performing sML model (SML2) underwent optimization through feature engineering (incorporating wavelet scattering for time-frequency analysis) and threshold adjustment. While computationally intensive, wavelet scattering enhanced frequency detail capture. Random kernel transformation (ROCKET) was also explored as a rapid time series analysis method.

The final algorithm (SML3) combined ROCKET, wavelet scattering features, and clinical covariates, with interaction coefficients from L1-regularized logistic regression and a multilayer perceptron, achieving 100% accuracy (AUC = 1.0) in the test set (Table 2). The model’s decision-making process was further elucidated through normalized mutual information analysis between biosignature representations and raw features, integrated via L2 and L1 classifiers. Feature visualization using ANOVA of wavelet transform-based Mel-scaled vibroacoustic biosignatures supported these results (Figure 2 and Figure 3).

Analysis to determine the minimal auscultation strategy identified a simplified 6-step anterior-only staircase procedure (30 s per site, total 3 min, starting at the right carotid while sitting or standing) as sufficient to maintain high performance metrics (sensitivity, specificity, accuracy, AUC) meeting pre-defined target product profiles (Figure 4).

## 4. Discussion

By recognizing the lungs as dynamic fluid pumps with predictable failure patterns, this vibroacoustics host-response framework enables earlier detection, accurate diagnostics, and scalable monitoring of respiratory diseases.

Noninvasive, point-of-need host response diagnostics offer a transformative, threat-agnostic approach to infectious disease detection by decoding the body’s immune and physiological responses rather than targeting specific pathogens. This paradigm enables earlier identification of infection, stratification of disease severity, and informed clinical decision-making—even in cases where the causative agent remains unknown or undetectable. By capturing real-time signatures of immune activation and infection trajectory, host response diagnostics provide a comprehensive view of patient status that transcends the limitations of traditional pathogen-based tests. Critically, they also potentially support global antimicrobial stewardship by distinguishing between viral and bacterial infections, helping reduce unnecessary antibiotic use. Scalable, rapid, and adaptable, this approach represents a major advance in preparedness for emerging infectious threats and in the delivery of precision infection management. This study marks the initial go/no-go milestone (see Section 4.2) in a broader effort to validate vibroacoustic biosignatures for noninvasive infectious disease diagnosis.

### 4.1. A Biophysics Hypothesis—Directed Respiratory Health Biomarker Testing Framework

A biophysics-informed framework for host response diagnostics leverages the body’s mechanical vibrations—from infrasonic (<20 Hz), audible, to ultrasonic (>20 kHz)—as a powerful lens into deep physiological function. This framework is unified by four core principles: tensegrity, fluid mechanics, mechanotransduction, and wave theory. In the lungs, tensegrity describes the architectural balance between continuous tension (elastin, fascia) and discontinuous compression (cartilage, alveoli), preserving airway patency and compliance. As inspired air moves through bronchial pathways, fluid mechanics governs flow dynamics, where changes in viscosity, obstruction, or inflammation create friction and shear stress. Mechanotransduction underpins these assessments by translating pressure and shear forces into cellular and systemic responses—essential for interpreting lung ultrasound findings in pneumonia [16,17], where fluid–structure interactions reveal disease-specific patterns. Similarly, vibrocardiography [18,19] and phonocardiography [20,21,22] capture the tension–compression dynamics of the heart and great vessels, translating micro-deformations into acoustic signals. In the GI system, vibroacoustic methods assess motility and recovery [23,24,25], mapping wave-transmitted tissue activity across complex viscoelastic networks. These interactions generate biomechanical vibrations—air–tissue friction, turbulence, and shifting compliance—which propagate as shear waves through the lungs, vasculature, and surrounding tissues. Across these domains, wave theory integrates signal propagation with tissue biomechanics, offering a unified lens to decode health and disease through mechanical listening, enabling detection of these subtle ripple-like “bending waves” on the skin. By capturing the body’s continuous audible and inaudible vibrational output, vibroacoustic sensors generate non-invasive, scalable, and interpretable data. AI algorithms can translate these biosignatures into early indicators of infection—detecting host response patterns specific to viral threats, before symptoms or pathogen identification.

As a public health tool, host-response vibroacoustic diagnostics redefine preparedness. They turn breath into data, data into foresight, and foresight into action—offering a passive, threat-agnostic early warning system capable of intercepting future pandemics before they take hold.

### 4.2. A Rigorous Fast-Fail Matched Case—Control Design for Biomarker Hypothesis Testing

A rigorous matched case–control study design offers an efficient and cost-effective framework for testing digital biomarker hypotheses by leveraging well-characterized participants with clear differentiation between disease and non-disease groups. This design eliminates the need for long-term follow-up, focusing instead on collecting high-quality data that rapidly generates actionable evidence [26,27,28]. Matching cases and controls on critical variables such as age, gender, and health status reduces confounding variables and increases the signal-to-noise ratio, enabling disease-specific features to emerge more clearly.

This proof-of-concept study employed a fast-fail matched case–control design to assess the feasibility of classifying COVID-19 using vibroacoustic biosignatures. Despite a small sample size of 15 matched pairs and 19,440 vibroacoustic samples, the approach enabled efficient evaluation within the study’s limited scope, demonstrating the potential for rapid, scalable diagnostic development.

Insights from this study, including optimized vibroacoustic biosignature effect-size, reduced auscultation sites, and shorter scan durations, demonstrate the potential of matched case–control designs to inform the design of larger confirmatory trials and accelerate the development of scalable, AI-driven digital biomarkers for global health challenges, see Table 3.

### 4.3. Establishing High-Quality Ground Truth Data for AI-Based COVID-19 Diagnostics

Establishing a robust foundation of “ground truth” data is critical for training and evaluating AI diagnostic algorithms. In this proof-of-concept pilot study, a matched case–control design was implemented to create a clinically relevant and unbiased dataset. Participants were acutely ill, hospitalized COVID-19 positive and negative individuals. COVID-19 status was confirmed using high-sensitivity Emergency Use Authorization (EUA) RT-PCR testing, the gold standard for diagnosis, ensuring accurate categorization and addressing common critiques of digital biomarker studies [31].

To ensure data integrity, recordings were conducted in controlled environments, minimizing variability from external noise or conditions. Vibroacoustic data were securely captured using dedicated hardware. Clinical data were meticulously curated to ensure traceability and quality. Vibroacoustics data were encrypted and compressed at the source to maintain compliance with ethical and regulatory standards. This rigorous process generated a high-confidence dataset, enabling the development of AI algorithms focused on disease-specific features rather than general illness indicators. Such precision is essential for producing reliable, meaningful diagnostics that can transition seamlessly into real-world clinical applications.

### 4.4. A Collaborative-Competition Framework for Curated Crowdsourcing of Cutting-Edge Algorithms

We implemented a collaborative-competition, “Collaboratition”, to identify useful diagnostic algorithms beyond those proposed by the technology development team. First, a collaborative phase encouraged diverse teams seeking access to novel curated datasets to innovate using a shared training dataset while exchanging methodology ideas. Next, a competitive phase provided a definitive performance benchmark where a neutral biostatistician evaluated each dockerized algorithm against a sequestered, blinded test set. In this way we could foster innovation while maintaining strict data integrity and evaluation standards.

Central to this framework was a federated Docker system, enabling secure access to codebases and datasets without transferring or duplicating large files. Data security and compliance were ensured, and risks of overfitting or inflated performance metrics were minimized by preventing overlap between training and testing datasets. A blinded cohort was reserved for final evaluation, providing unbiased assessment and robust validation of algorithm performance.

Teams worked independently to optimize diverse algorithmic approaches while periodically sharing intermediate results and methodologies. This collaborative exchange enriched problem-solving strategies, avoided local optimization pitfalls, and promoted continuous improvement through varied techniques. The use of AI-generalized blinded test data approach further ensured algorithms were evaluated on entirely unseen datasets, offering an objective measure of their real-world applicability.

By combining competitive excellence with curated collaboration in a cloud-enabled environment, this framework accelerated innovation, facilitated exploration of cutting-edge algorithms, and established a replicable model for efficiently addressing complex biomarker discovery challenges.

### 4.5. Study Limitations and Future Steps

This proof-of-concept study focused on assessing the feasibility of detecting a COVID-19 host-response vibroacoustic biosignature in hospitalized patients. While technically promising, several limitations must be acknowledged.

First, the study population included only symptomatic, hospitalized individuals. The study did not include individuals with asymptomatic SARS-CoV-2 infection or other viral respiratory illnesses by design. This was intended to be a go/no-go initial study—had we not been able to distinguish between those hospitalized with symptomatic COVID and those without respiratory disease using the device and our best algorithms, we would have ceased development of the tool for this indication. We have subsequently embarked on trials amongst those with asymptomatic disease and with other viral illnesses. Second, the small sample size (*n* = 35) raises concerns about statistical power and overfitting, particularly given the model’s perfect performance when combining vibroacoustics and host covariates (AUC = 1.0) and reduced performance with an unoptimized vibroacoustics only model (AUC = 0.752). Although the optimized model (SML3) achieved high accuracy, these findings require validation in larger, independent cohorts.

Third, confounding factors may not have been fully controlled, and BMI, while matched, is an imperfect proxy for obesity and body composition. Fourth, while grounded in biophysical theory, our approach lacks direct experimental or molecular validation of the proposed mechanotransductive mechanisms, relying instead on machine learning–based pattern discovery.

Lastly, the initial protocol, which generated ~54 min of data per participant across multiple postures and sites, was designed for signal discovery. From this, we derived a simplified 6-site anterior chest protocol suitable for clinical use. However, signal measurements were limited to two sessions over 72 h, leaving long-term stability and responsiveness to disease progression an area for future exploration.

While not designed for cost analysis, we note that passive vibroacoustic sensing may offer lower per-use costs than disposable tests due to its reusable, non-contact design. Future work will explore its cost-effectiveness, usability, and real-world deployment potential.

## 5. Conclusions

This proof-of-concept study leveraged a high-fidelity RT-PCR ground truth dataset (PPA > 95%) to reconceptualize the lungs as a finely tuned musical instrument—where subtle, often inaudible vibrations encode rich diagnostic information. Importantly, this approach bypasses vocal artifacts and instead captures deep physiological signals rooted in wave theory and mechanotransduction, revealing a novel, actionable dimension of disease detection. Using an embedded AI stethoscope capable of capturing infrasonic to ultrasonic frequencies, we identified noninvasive host-response vibroacoustic biosignatures that, when combined with clinical data, achieved near-perfect sensitivity and specificity for COVID-19 within a rigorously matched case–control design. By design, this study evaluated a highly selected population with clear separation between cases and controls, providing an essential, controlled environment for initial signal discovery and validation. The findings establish a foundational fast-fail, high-evidence framework for evaluating novel diagnostics—supporting scalable, affordable, and globally deployable strategies for infectious disease detection.

While the results in this curated setting are promising, the true challenge lies ahead: validating this approach in more complex, real-world populations, including asymptomatic carriers, patients with co-infections or comorbid respiratory conditions, and diverse cohorts across geographies and healthcare settings. As we transition from a well-controlled proof-of-concept to broader clinical validation, this work positions passive vibroacoustic sensing as a compelling, threat-agnostic paradigm for pre-symptomatic screening, early intervention, and convenient monitoring. As the world prepares for future pandemics, the ability to noninvasively hear, feel, see, and interpret host response signatures of infection offers an urgently needed addition to the global diagnostic toolkit.

## Figures and Tables

**Figure 1 viruses-17-00943-f001:**
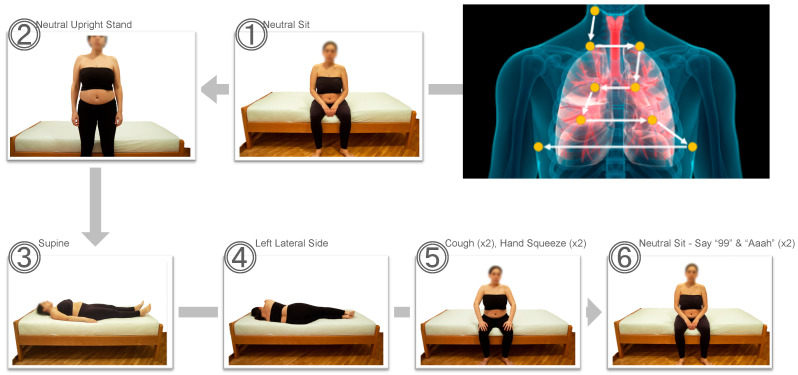
A 9-step staircase (white arrows) vibroacoustic scan procedure at different auscultation points (yellow circles) and six patient posture/positions. Patients not able to sit or stand were not excluded from the pilot study. They were allowed to participate in the parts of the staircase exam that they could complete. Cough, hand squeeze and “99, Aaah” vocalization was performed in a seated position.

**Figure 2 viruses-17-00943-f002:**
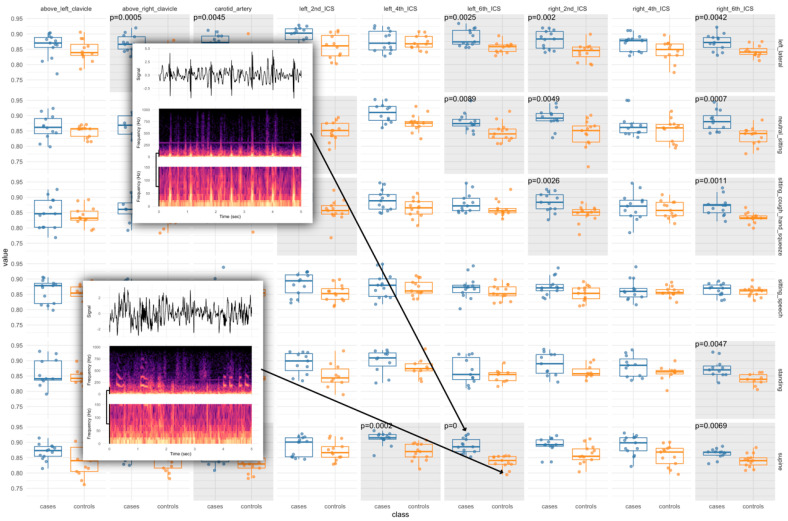
Visual comparisons of Mel-scale based features of spectrograms of matched case–control pair samples. Blue indicates features derived from cases, and orange indicates features from controls. Gray-shadowed regions highlight case-control pair wavelet transform-based Mel-scaled features that were found to be statistically significantly different between the two groups, as determined by ANOVA.

**Figure 3 viruses-17-00943-f003:**
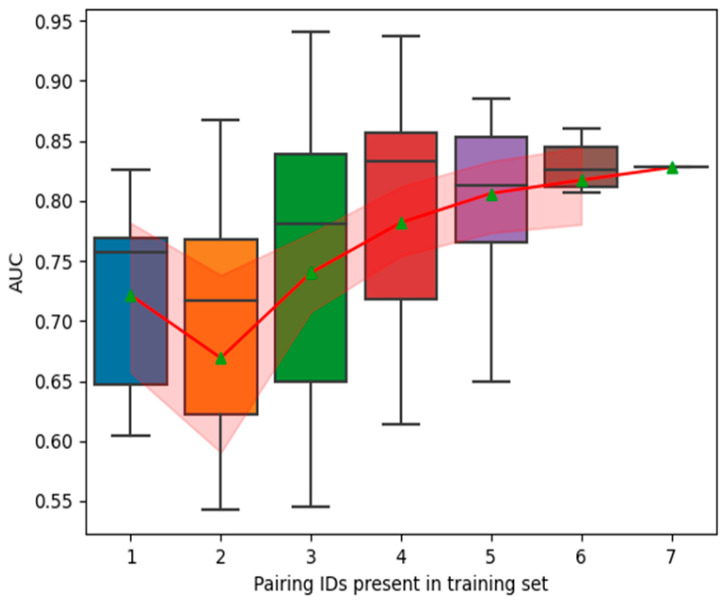
The SML2 Learning Curve increased towards an AUC asymptote (AUC~0.85%) with the size of the training set pairings. The boxplot shows median (line), interquartile range (box range) and whole distribution (whiskers) without outliers. The line plot connects means (triangles). The 95% mean confidence interval of the line plot is shaded. We subsequently used this inverse power law learning curve framework to estimate clinical validation trial sample sizes required to achieve/exceed target product profile predictive performance.

**Figure 4 viruses-17-00943-f004:**
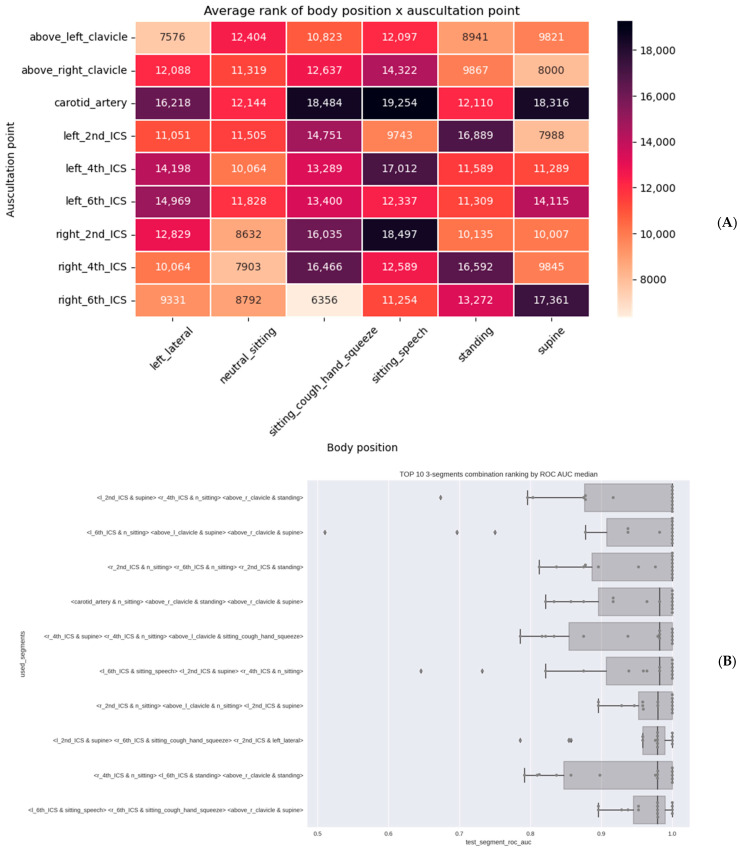
(**A**) Heatmap illustrating the informativeness of auscultation point-body position audible and inaudible vibroacoustic data segment types (the higher the value, the higher the importance of vibroacoustic segments, auscultation points and body positions). (**B**) Top ten 3-tuples of segments (of all possible 3-tuples) according to the median ROC AUC on session level. We considered all 3-tuples of segments; given 9 auscultation points and 6 body positions, and thus 54 segments, there are 24,804 such unique 3-tuples.

**Table 1 viruses-17-00943-t001:** Sociodemographic and Clinical Characteristics of Case Patients and Control Subjects of Matched Case-Control Pilot Study.

	All Patients (*n* = 35)	COVID-19 Negative Group (*n* = 16)	COVID-19 Positive Group (*n* = 19)
Demographic variable	Median (interquartile range)		
Age (years)	48.6 (30–78)	49.0 (30–70)	48.3 (32–78)
Body Mass Index (BMI)	32.4 (19.2–44.3)	32.4 (19.2–44.3)	32.4 (21.2–43.6)
Demographic variable	N (%)		
Sex			
Male	15 (43)	7 (44)	8 (42)
Female	20 (57)	9 (56)	11 (58)
Race			
Asian or Asian American	1 (3)	0 (0)	1 (5)
Black or African American	17 (49)	8 (50)	9 (47)
White	16 (46)	8 (50)	8 (42)
Multi race	1 (3)	0 (0)	1 (5)
Ethnicity			
Hispanic origin	3 (9)	0 (0)	3 (16)
Not of Hispanic origin	32 (91)	16 (100)	16 (84)
Smoking history			
Use of tobacco currently or within last 3 months	3 (9)	1 (2)	2 (11)
No current tobacco use	32 (91)	15 (93)	17 (89)
Laboratory test	Median (interquartile range)		
White blood cell (K/cu mm)	7.60 (2.21–20.37)	8.67 (2.59–20.37)	6.69 (2.21–12.55)
Absolute lymphocyte count (K/cu mm)	1.67 (0.33–5.22)	2.19 (0.71–5.22)	1.24 (0.33–3.26)
C-Reactive Protein (mg/L) *	3.88 (0.10–18.50)	2.04 (0.10–9.30)	5.42 (0.20–18.50)
Erythrocyte sedimentation rate (mm/h)	51.25 (4.00–130.00) (*n* = 28)	54.91 (4.00–130.00) (*n* = 11)	48.88 (6.00–88.00) (*n* = 17)
Interleukin 6 (pg/mL)	34.37 (1.20–372.00) (*n* = 31)	20.17 (2.70–100.00) (*n* = 13)	44.62 (1.20–372.00) (*n* = 18)
D-dimer (mg/L)	1.38 (0.28–8.79) (*n* = 25)	2.26 (0.31–8.79) (*n* = 7)	1.04 (0.28–3.86) (*n* = 18)
Lactate dehydrogenate (U/L)	260.37 (123.00–576.00) (*n* = 30)	220.69 (123.00–339.00) (*n* = 13)	290.71 (141.00–576.00) (*n* = 17)
Ferritin (ng/mL)	833.51 (20.00–10,223.00)	590.19 (62.00–5286.00)	1038.42 (20.00–10,223.00)
COVID-19 Ab IgG (AU/mL)	1.40 (0.00–11.90) (*n* = 30)	0.51 (0.00–6.59) (*n* = 13)	2.08 (0.00–11.90) (*n* = 17)
COVID-19 AB IgA (AU/mL)	3.18 (0.00–40.04) (*n* = 30)	0.00 (0.00–0.00) (*n* = 13)	5.62 (0.00–40.04) (*n* = 17)
Medical history	N (%)		
Diabetes	13 (37)	8 (50)	5 (26)
Hypertension	19 (54)	9 (56)	10 (53)
Hyperlipidemia	11 (31)	4 (25)	7 (37)
Coronary artery disease	3 (9)	1 (6)	2 (11)
Heart failure	3 (9)	2 (13)	1 (5)
Arrhythmia	1 (3)	1 (6)	0 (0)
Implantable devices in chest	2 (6)	1 (6)	1 (5)
Mechanical heart valve	0 (0)	0 (0)	0 (0)
Pacemaker	0 (0)	0 (0)	0 (0)
Peripheral artery disease	1 (3)	1 (6)	0 (0)
Cerebrovascular disease	5 (14)	4 (25)	1 (5)
Chronic lung disease	4 (11)	1 (6)	3 (16)
HIV	4 (11)	1 (6)	3 (16)
Tuberculosis	0 (0)	0 (0)	0 (0)
Liver disease	2 (6)	0 (0)	2 (11)
Kidney disease	7 (20)	2 (13)	5 (26)
Pregnancy	1 (3)	0 (0)	1 (5)
Cancer	0 (0)	0 (0)	0 (0)
Dialysis	0 (0)	0 (0)	0 (0)
Other immunodeficiency	3 (9)	1 (6)	2 (11)

Definition of abbreviations: BMI = mass (kg)/height^2^ (m^2^); HIV = human immunodeficiency virus; AU/mL = arbitrary units (AU) per milliliter (mL). * If value C-Reactive Protein (mg/L) < LLN, the mean was calculated using 0.3 mg/L.

**Table 2 viruses-17-00943-t002:** Blinded out-of-sample accuracy of SML and State-of-the-Art DL models on small training cohorts (‘*n*-sense’) despite availability of extended durations of vibroacoustic data collection (‘*t*-sense’) within the limited-size matched case–control pilot study.

	SML1	SML2	SML3 ^#^	DL1	DL2
True Positive	11	13	19	11	10
True Negative	15	14	16	12	17
False Positive	2	3	0	5	0
False Negative	7	5	0	9	8
Accuracy	74.3%	77.1%	100.0%	62.2%	77.1%
Sensitivity	61.1%	72.2%	100.0%	55.0%	55.6%
Precision	84.6%	81.3%	100.0%	68.8%	100%
Specificity	88.2%	82.4%	100.0%	70.6%	100%
F-score	71.0%	76.5%	100.0%	61.1%	71.4%
AUC	78.4%	83.7%	100.0%	64.7%	75.2%

SML1, SML2, DL1, and DL2 were trained on matched case–control study data and tested in ‘blinded contest mode’ whereby trained algorithms were uploaded via Docker for testing against unseen data behind a firewall. SML3 ^#^ was developed by redefining training/test sets, optimizing SML2 via feature engineering and sensitivity/specificity thresholding. Definition of abbreviations: F-score = harmonic mean of a system’s precision and recall values. It is calculated by the following formula: 2 × [(Precision × Recall)/(Precision + Recall)]. AUC = area under the receiver operating characteristic curve; SML = structural machine learning, DL = State-of-the-Art deep neural networks/deep learning.

**Table 3 viruses-17-00943-t003:** Comparative performance of the proposed vibroacoustic host-response biosignature approach versus current State-of-the-Art diagnostic methods. This table highlights the accuracy of our novel, noninvasive strategy for decoding host-response signals using vibroacoustic sensing. By situating our findings alongside conventional pathogen-based and biomarker-driven approaches, the comparison underscores the potential of this method to advance the field of biosignature discovery—particularly in enabling early, threat-agnostic, and scalable infection diagnostics.

Test Set	AUC (95% CI)	Sensitivity (95% CI)	Specificity (95% CI)	Reference
Temperature assessment	n/a	(0.00–0.23)	(0.90–1.00)	[29]
Symptom Assessment	n/a	(0.00–0.60)	(0.66–1.0)	[29]
Symptom plus Temp Assessment	n/a	(0.12–0.69)	(0.90–1.00)	[29]
Antigen Test vs. symptomatic individuals	n/a	0.739 (0.684–0.79)	0.984–0.997	[30]
Antigen Test vs. asymptomatic individuals	n/a	0.402 (0.215–0.622)	0.984–0.997	[30]
SML2 (device only)	0.837 (0.673–0.921)	0.722 (0.578–0.872)	0.813 (0.639–0.944)	This manuscript
SML3 (device + clinical covariate data)	1.0 (0.94–1.0)	1.0 (0.818–1.0)	0.996 (0.845–1.0)	This manuscript

## Data Availability

The anonymized data, upon which this report is based, are too large to archive or to transfer and will remain at rest. Further information about the data and conditions for access will be made available upon publication at viruses_dataaccess@level42.ai.

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
