# Peer review of "Evidence Generation for a Host-Response Biosignature of Respiratory Disease"

_viruses, 2025, doi:10.3390/v17070943_

Round 1
Reviewer 1 Report
Comments and Suggestions for Authors
Thank you for the opportunity to review this compelling article by Kelly Dooley et al. The study introduces an original and innovative technology grounded in detailed physics for detecting vibroacoustic signatures in pulmonary (infra)sounds associated with viral infections. This case-control study is meticulously designed and appears to have been conducted with a high degree of rigor. While the data analysis employs machine learning techniques that may be unfamiliar to some medical researchers, the results are clearly and effectively presented. I have no significant concerns.
Author Response
Thank you for the opportunity to review this compelling article by Kelly Dooley et al. The study introduces an original and innovative technology grounded in detailed physics for detecting vibroacoustic signatures in pulmonary (infra)sounds associated with viral infections. This case-control study is meticulously designed and appears to have been conducted with a high degree of rigor. While the data analysis employs machine learning techniques that may be unfamiliar to some medical researchers, the results are clearly and effectively presented. I have no significant concerns.
Response: We thank the reviewer for their time, interest, and effort.
Reviewer 2 Report
Comments and Suggestions for Authors
The aim of this study is to identify reproducible vibroacoustic patterns associated with host response to infection, focusing on subtle alterations in compressible air-fluidynamics and biomechanical tissue properties. The study suggests that early infection in COVID-19 patients alters lung tension, wave propagation, and flow dynamics through subaudible vibroacoustic signals, which can be detected through machine learning models from anterior chest recordings.
While the manuscript addresses a clinically relevant and timely topic, it requires further clarification of methodology, improved data presentation.
- In section 3.2, “Algorithm Training and Testing” This section requires further elaboration. Specifically, the manuscript should clearly state which machine learning (ML) and deep learning (DL) models were used (e.g., Random Forest, XGBoost, SVM, CNN, LSTM). Additionally, important training details—such as number of epochs, batch size, and optimizer type—should be included to enhance reproducibility and transparency.
- In section 4.4, The concept of a "Collaborative Competition Framework" is promising, but it is insufficiently described. The manuscript should provide more detail on how this framework was defined, structured, and implemented, including how collaboration and competition were managed and how it influenced the study outcomes.
- In the discussion section, A comparison table or contrasting the proposed approach with state-of-the-art methods should also be added to better contextualize the contribution in the area of context of advancing biosignature discovery.
Author Response
In section 3.2, “Algorithm Training and Testing” This section requires further elaboration. Specifically, the manuscript should clearly state which machine learning (ML) and deep learning (DL) models were used (e.g., Random Forest, XGBoost, SVM, CNN, LSTM). Additionally, important training details—such as number of epochs, batch size, and optimizer type—should be included to enhance reproducibility and transparency.
Response: Thank you for your careful reading of the methodology and for your suggestion. We have modified Section 3.2 (p7, line 200) to provide more details about the approaches we used. It has been rewritten as follows: Analytic approaches included deep learning (DL) and model-based algorithms (structural machine learning, sML) using domain-engineered features. The DL approach utilized three architectures: Long Short-Term Memory (LSTM), Convolutional LSTM (ConvLSTM), and transfer learning models based on the YAMNet pre-trained audio classifier. The sML method employed an ensemble of diverse classifiers—including logistic regression, random forest, and support vector machines (SVM)—to combine their individual predictions into a single, aggregate probability. For training, the models were trained for 100 epochs using a mini-batch Stochastic Gradient Descent (SGD) optimizer with a batch size of 32. On the training and validation datasets, both vibroacoustics only DL and sML models achieved an area under the receiver operating characteristic curve (AUC) of 1.0. However, on the blinded test set, the best DL model yielded an AUC of 0.752, while the unoptimized sML model achieved an AUC of 0.837 (Table 2). The drop in training-vs.-test performance provides an indirect measure of model over-fitting.
In section 4.4, The concept of a "Collaborative-Competition Framework" is promising, but it is insufficiently described. The manuscript should provide more detail on how this framework was defined, structured, and implemented, including how collaboration and competition were managed and how it influenced the study outcomes.
Response: We added (p14, line 350) introductory information, as follows: We implemented a collaborative-competition, “Collaboratition”, to identify useful diagnostic algorithms beyond those proposed by the technology development team. First, a collaborative phase encouraged diverse teams seeking access to novel curated datasets to innovate using a shared training dataset while exchanging methodology ideas. Next, a competitive phase provided a definitive performance benchmark where a neutral biostatistician evaluated each dockerized algorithm against a sequestered, blinded test set.
In the discussion section, A comparison table or contrasting the proposed approach with state-of-the-art methods should also be added to better contextualize the contribution in the area of context of advancing biosignature discovery.
Response: Yes, great idea, thank you. We have inserted (p13, line 322) the following table into the manuscript. In brief, you can see that with impulse UNA and clinical information, sensitivity and specificity approach that of the antigen test in individuals with symptoms. Where antigen tests are not readily available (or are expensive for single use), the e-stethoscope serves as a useful biomarker.
|
Test set |
AUC (95% CI) |
Sensitivity (95% CI) |
Specificity (95% CI) |
Reference |
|
Temperature assessment |
n/a |
(0.00-0.23) |
(0.90-1.00) |
32 |
|
Symptom Assessment |
n/a |
(0.00-0.60) |
(0.66-1.0) |
32 |
|
Symptom plus Temp Assessment |
n/a |
(0.12-0.69) |
(0.90-1.00) |
32 |
|
Antigen Test vs. symptomatic individuals |
n/a |
0.739(0.684-0.79) |
0.984-0.997 |
33 |
|
Antigen Test vs. asymptomatic individuals |
n/a |
0.402(0.215-0.622) |
0.984-0.997 |
33 |
|
SML2 (device only) |
0.837 (0.673-0.921) |
0.722 (0.578-0.872) |
0.813 (0.639-0.944) |
This manuscript |
|
SML3 (device + clinical covariate data) |
1.0 (0.94–1.0) |
1.0 (.818–1.0) |
0.996 (0.845–1.0) |
This manuscript |
- Dinnes J, Sharma P, Berhane S, van Wyk SS, Nyaaba N, Domen J, Taylor M, Cunningham J, Davenport C, Dittrich S, Emperador D, Hooft L, Leeflang MM, McInnes MD, Spijker R, Verbakel JY, Takwoingi Y, Taylor-Phillips S, Van den Bruel A, Deeks JJ; Cochrane COVID-19 Diagnostic Test Accuracy Group. Rapid, point-of-care antigen tests for diagnosis of SARS-CoV-2 infection. Cochrane Database Syst Rev. 2022 Jul 22;7(7):CD013705.
- Arshadi Maniya, Fardsanei Fatemeh, Deihim Behnaz, Farshadzadeh Zahra, Nikkhahi Farhad, Khalili Farima, Sotgiu Giovanni, Shahidi Bonjar Amir Hashem, Centis Rosella, Migliori Giovanni Battista, Nasiri Mohammad Javad, Mirsaeidi Mehdi. Diagnostic Accuracy of Rapid Antigen Tests for COVID-19 Detection: A Systematic Review With Meta-analysis. Frontiers in Medicine. 9. 2022.
Reviewer 3 Report
Comments and Suggestions for Authors
The proposed manuscript describes the results of a study aimed at identifying reproducible vibroacoustic patterns associated with host response to infection, focusing on subtle alterations in compressible air-fluid dynamics and biomechanical tissue properties.
The authors apply matched case-control study, clinical laboratory tests, collection and processing of vibroacoustic data of participants with confirmed SARS-CoV-2 infection and matched controls. Machine learning models were applied and optimized to identify host-response biosignatures from anterior chest recordings.
Preliminaries to the research area are provided. In particular, the authors review recent methods for diagnostics of infectious diseases mentioning that most of them remain pathogen-focused, often missing the earliest phase of infection. The need of a virus-agnostic, host-based diagnostic capable of detecting pre-symptomatic responses to viral intrusion is emphasized.
The proposed methodology and the obtained results are carefully described. Details about the subjects, experimental design, data collection, and pre-processing are described in detail. The idea of machine learning algorithms as well as their optimization procedures are explained.
The results obtained are presented carefully. They are illustrated by many figures and are compared with other methodologies.
The presentation of the main results is clear and comprehensive. From a formal point of view, all the contents seems to be correct. The results are valuable and worthy of being published taking into account their potential applications in diagnostics and clinical practice.
Minor revisions are suggested to improve the quality of the exposition:
p.5, line 140: The abbreviation BMI is given twice.
p.7, line 213: It should be “SML1” instead of “SM1”.
p.9, line 235: The quality and readability of Fig. 4 (B) is bad, probably (A) and (B) could be separated and enlarged.
Author Response
p.5, line 140: The abbreviation BMI is given twice.
Response: Thank you. This has been corrected.
p.7, line 213: It should be “SML1” instead of “SM1”.
Response: Thank you. This has been corrected.
p.9, line 235: The quality and readability of Fig. 4 (B) is bad, probably (A) and (B) could be separated and enlarged.
Response: Thank you. We have created separate figures.
Reviewer 4 Report
Comments and Suggestions for Authors
This study proposes a non-invasive approach based on vibroacoustic biomarkers to detect host immune responses in COVID-19 patients. The hypothesis is that pulmonary biomechanical alterations (e.g., changes in tissue tension and airflow dynamics) induced by infection manifest as micro-vibrational signals. By employing machine learning to analyze chest vibration signals across infrasonic (<20 Hz), audible (20–20,000 Hz), and ultrasonic (>20,000 Hz) frequency ranges, the research team successfully extracted COVID-19-specific biomarkers.
There are several issues to address
- The study primarily focused on distinguishing COVID-19 patients from non-COVID-19 controls but did not thoroughly investigate whether these vibroacoustic biomarkers can effectively differentiate other respiratory conditions (e.g., influenza, bacterial pneumonia, chronic obstructive pulmonary disease). Since different respiratory diseases may induce similar pulmonary mechanical changes and host responses, the clinical utility of this method could be limited if it lacks specificity, potentially leading to misdiagnosis or missed cases. The study population consisted of hospitalized patients, excluding those on mechanical ventilation, which may limit generalizability to mild or asymptomatic cases and community settings. Additionally, the absence of other respiratory viral infections (e.g., influenza) in the analysis prevents assessment of broader applicability. Studies should explore the method’s performance in monitoring mild or asymptomatic individuals in community settings. A direct comparison with traditional methods (e.g., chest CT, conventional auscultation) or emerging host-response biomarkers (e.g., transcriptomics) is also needed to evaluate relative advantages.
- The model demonstrated perfect performance on the training set (AUC=1.0) but showed reduced efficacy on the unoptimized test set (DL model AUC=0.752). Although the final optimized model (SML3) achieved 100% accuracy on the test set, the small sample size raises concerns about overfitting. The authors should explain the issues associated with model overfitting.
- Despite matching for age, BMI, and smoking history, other potential confounders (e.g., underlying lung conditions, medication effects) may not have been fully controlled. For exmaple, chronic pulmonary diseases or obesity could independently influence vibroacoustic signals.
- While the study presents biophysics-based hypotheses (e.g., tensegrity, fluid dynamics, mechanotransduction, and wave theory), it lacks in-depth experimental validation or molecular-level explanations for the relationship between vibroacoustic signals, host immune responses, and pulmonary pathophysiological changes. This gap somewhat undermines the scientific rigor and credibility of the method and hinders further optimization and application. The assumption that lung biomechanical changes generate specific signals was not directly tested through experiments (e.g., tissue mechanics measurements or cellular mechanism studies), relying instead on indirect machine learning correlations.
- The dataset was not made publicly available due to its large size, which may impede independent validation. Additionally, some authors are affiliated with the funding company (Level 42 AI), raising potential conflicts of interest. The study did not discuss equipment costs, training requirements, or cost-effectiveness comparisons with other screening tools (e.g., rapid antigen tests). I recommend the authors add the discussion about the cost across different screening tools.
- Although the study ultimately proposed a simplified 6-step anterior chest detection protocol, the original data collection involved 9 auscultation points, 6 body postures/positions, and 30-second recordings per point, resulting in a massive dataset (~54 minutes of vibroacoustic data per participant). This complexity, along with the need for advanced signal processing and machine learning algorithms, may hinder scalability in resource-limited or primary care settings. The authors should acknowledge this limitation in the manuscript.
- The study only assessed short-term signal variations (with two visits 24–72 hours apart) and lacked long-term follow-up, leaving the long-term validity and stability of these biomarkers unknown for disease progression, treatment monitoring, or recovery assessment. For example, whether the signals change predictably with clinical improvement or deterioration, or whether the method can dynamically track disease course, remains unaddressed—critical gaps for clinical applicability. The authors should acknowledge this limitation in the manuscrip
- The study population comprised only hospitalized symptomatic patients, whereas one of the stated goals was early or pre-symptomatic detection. The absence of asymptomatic or early-stage pre-symptomatic cases limits validation of the method’s effectiveness for true early detection, restricting its potential for disease prevention and control. The authors should acknowledge this limitation in the manuscrip
- With only 19 COVID-19 hospitalized patients and 16 matched controls, the small sample size may compromise statistical power and fail to represent real-world diversity (e.g., across age, sex, comorbidities, disease severity), limiting generalizability. The authors should explicitly state this limitation.
Author Response
The study primarily focused on distinguishing COVID-19 patients from non-COVID-19 controls but did not thoroughly investigate whether these vibroacoustic biomarkers can effectively differentiate other respiratory conditions (e.g., influenza, bacterial pneumonia, chronic obstructive pulmonary disease). Since different respiratory diseases may induce similar pulmonary mechanical changes and host responses, the clinical utility of this method could be limited if it lacks specificity, potentially leading to misdiagnosis or missed cases. The study population consisted of hospitalized patients, excluding those on mechanical ventilation, which may limit generalizability to mild or asymptomatic cases and community settings. Additionally, the absence of other respiratory viral infections (e.g., influenza) in the analysis prevents assessment of broader applicability. Studies should explore the method’s performance in monitoring mild or asymptomatic individuals in community settings. A direct comparison with traditional methods (e.g., chest CT, conventional auscultation) or emerging host-response biomarkers (e.g., transcriptomics) is also needed to evaluate relative advantages.
Response: Exactly! This was a proof-of-concept study at the beginning of the COVID-19 pandemic aimed at taking the first step in development of a non-invasive test that could be used to detect COVID-19. It was designed to look at the very black and white scenario where patients either had symptomatic COVID-19 or they did not have any type of respiratory disease. If we could not use e-stethoscope-generated data to discriminate amongst these two scenarios, we would have stopped development of the device for this indication. Following this positive go/no-go study, we then embarked on diagnostic trials among those with asymptomatic SARS-CoV-2 and with other respiratory pathogens.
The primary aim of this pilot study was not to produce a deployable classifier, but rather to explore whether a host-response vibroacoustic biofield signature of COVID-19 infection could be detected and characterized in a diverse cohort of confirmed COVID-19–positive participants. The use of machine learning in this context was intended as a discovery tool to assess signal presence and separability, rather than to generalize predictive performance to broader populations.
We have modified the manuscript (Abstract- p1, lines 21, 30, 34; p2, line 80; p 12, line 259, and p16, line 423) to caveat the conclusions of the presented proof-of-concept study.
The model demonstrated perfect performance on the training set (AUC=1.0) but showed reduced efficacy on the unoptimized test set (DL model AUC=0.752). Although the final optimized model (SML3) achieved 100% accuracy on the test set, the small sample size raises concerns about overfitting. The authors should explain the issues associated with model overfitting.
Response: We thank the reviewer for raising the important issue of overfitting. We fully acknowledge that overfitting is a potential risk in any pilot study involving small sample sizes. In our study, the final SML3 (vibroacoustics + host variables) model demonstrated perfect performance (AUC=1.0) on the training, validation, and test sets when clinical covariates were included, while initial evaluation of the unoptimized (vibroacoustics only) DL model yielded an AUC of 0.752 and the unoptimized SML2 model achieved an AUC of 0.837 (Table 2).
To address overfitting concerns, we designed a collaborative-competition approach across multiple teams using very different modeling approaches with each implementing cross-validation and independent test set evaluations, and we report both unoptimized and optimized model performances to provide transparency. In addition test data were withheld and only available to an independent unblinded statistician. We are the only group that we know of that consistently goes this far in putting measures in place to assess overfitting. However, we agree that external validation in larger, independent cohorts is essential to confirm generalizability. Since this pilot, we have performed studies with expanded sample sizes, COVID-19, RSV, flu exposed cohorts, and more rigorous external validation frameworks are planned to address these limitations and further evaluate the robustness of the identified biosignatures.
We have modified the manuscript (p7, line 212 and p15, 384) to callout and quantify model overfitting.
Despite matching for age, BMI, and smoking history, other potential confounders (e.g., underlying lung conditions, medication effects) may not have been fully controlled. For example, chronic pulmonary diseases or obesity could independently influence vibroacoustic signals.
Response: We appreciate the reviewer’s observation regarding potential residual confounders such as chronic pulmonary disease and medication effects. We agree that, despite matching participants for age, BMI, and smoking history, other underlying conditions—particularly those affecting pulmonary function—may not have been fully controlled in this pilot study.
We also recognize the limitations of BMI as a proxy for obesity. While BMI is widely used in research to stratify or control for obesity-related effects, it does not directly measure body fat or account for inter-individual variation in body composition, such as muscle mass, visceral fat distribution, or influences of race, ethnicity, and age. As such, its use introduces known limitations when evaluating physiological signal variability.
That said, the potential influence of obesity on signal quality differs between imaging and vibroacoustic modalities. Specifically, in ultrasound imaging, excess adiposity is known to degrade signal quality due to increased tissue thickness and attenuation of high-frequency sound waves, which impairs beam penetration and image clarity. In contrast, our approach uses passive vibroacoustic sensing, which captures low-frequency signals—including infrasonic and subaudible components—via body-coupled resonance. These frequencies propagate more efficiently through media such as fat and water than through air, making them less susceptible to the signal degradation commonly seen in ultrasound. Therefore, although body composition may still influence signal characteristics, passive vibroacoustics are relatively robust to the attenuation effects associated with obesity.
Future studies will incorporate additional clinical covariates such as underlying lung pathology, medication use, and refined measures of body composition (e.g., DEXA, impedance) to further disentangle confounding variables and validate the specificity of observed vibroacoustic biosignatures.
We have described the potential for unmeasured confounders in the limitations (p15, line 390) section of the manuscript.
While the study presents biophysics-based hypotheses (e.g., tensegrity, fluid dynamics, mechanotransduction, and wave theory), it lacks in-depth experimental validation or molecular-level explanations for the relationship between vibroacoustic signals, host immune responses, and pulmonary pathophysiological changes. This gap somewhat undermines the scientific rigor and credibility of the method and hinders further optimization and application.
Response: We appreciate the reviewer’s feedback regarding the desirability of deeper experimental validation and molecular-level mechanistic insights. We agree that the proposed relationship between vibroacoustic signals, host immune responses, and pulmonary pathophysiological changes is currently only supported through biophysics-informed hypotheses and indirect validation via machine learning. This is a recognized limitation of the present vibroacoustics host-response proof-of-concept study.
That said, it is important to emphasize that many widely accepted medical imaging modalities—including MRI, CT, and PET—are based on the principles of wave propagation and vibration, albeit through different physical mechanisms. Our approach similarly relies on foundational concepts in wave theory, mechanotransduction, and structural resonance, but it uniquely leverages passive acoustic emissions (biophony) as a novel host-response signal domain. While these signals differ from conventional imaging frequencies (e.g., magnetic or ionizing radiation), they are nonetheless governed by well-established physical laws that underpin biological and mechanical systems.
As a small pilot study, our aim was appropriately narrow: to determine whether distinguishable vibroacoustic biosignatures of COVID-19 infection could be detected in asymptomatic and symptomatic well-characterized individuals. The novelty of our work lies in showing that these vibroacoustic signatures likely reflect an aggregated host response—arising from structural, cellular, and functional alterations within the pulmonary and systemic compartments—even in the absence of targeted molecular assays or tissue biomechanics data. This discovery-based approach provides a foundation upon which future studies can build by incorporating direct experimental models, including tissue-level mechanical measurements, biofluid rheology, and cellular-level mechanotransductive pathway analysis.
We view this work as a necessary first step in establishing a new class of host physiological biosignatures and agree that further validation and mechanistic exploration will be critical for optimization, translational application, and broader scientific acceptance. We describe the types of next sets of experimental data that will help to advance this science in the manuscript.
We have added a statement to the manuscript (p15, line 388) acknowledging the Reviewer’s feedback.
The dataset was not made publicly available due to its large size, which may impede independent validation. Additionally, some authors are affiliated with the funding company (Level 42 AI), raising potential conflicts of interest. The study did not discuss equipment costs, training requirements, or cost-effectiveness comparisons with other screening tools (e.g., rapid antigen tests). I recommend the authors add the discussion about the cost across different screening tools.
Response: Due to the large size and complexity of the raw vibroacoustic recordings, direct inclusion within the manuscript was not feasible. However, we are committed to transparency and reproducibility. Upon acceptance of the manuscript, we will provide access to a secure online repository for general review and plan to offer a de-identified data download tool for qualified researchers to facilitate independent analysis and validation.
Second, we acknowledge that several authors are affiliated with Level 42 AI, the developer of the underlying technology. To mitigate any perceived conflict of interest, we followed best practices in declaring all affiliations and funding sources. Additionally, in line with the editorial team’s guidance, we intentionally focused this manuscript on the scientific characterization of host-response vibroacoustic biosignatures, rather than on the technical specifications, cost structure, or commercial details of the hardware, so as to avoid any perception of promotional intent.
Nonetheless, we agree that cost considerations are important for practical deployment. While a detailed cost-effectiveness analysis was outside the scope of this pilot study, we note that passive vibroacoustic sensing offers several potential advantages over current COVID-19 screening tools. These include non-invasive and contactless operation, real-time data capture, minimal need for consumables, and low marginal cost per use once deployed. In contrast to rapid antigen tests—which require recurrent purchase, physical handling, and biochemical reagents—our approach is reusable, scalable, and potentially cost-effective in high-throughput or resource-limited environments. Training requirements are minimal, as the system is designed for automated acquisition and AI-based interpretation, reducing the need for specialized personnel.
Future studies will more comprehensively evaluate cost, usability, and integration pathways relative to existing screening modalities. We appreciate the reviewer’s suggestion and will incorporate a brief summary of these considerations into the revised discussion.
We have added a statement to the manuscript (p15, line 407) acknowledging the Reviewer’s feedback.
Although the study ultimately proposed a simplified 6-step anterior chest detection protocol, the original data collection involved 9 auscultation points, 6 body postures/positions, and 30-second recordings per point, resulting in a massive dataset (~54 minutes of vibroacoustic data per participant). This complexity, along with the need for advanced signal processing and machine learning algorithms, may hinder scalability in resource-limited or primary care settings. The authors should acknowledge this limitation in the manuscript.
Response: We thank the reviewer for highlighting important considerations regarding the complexity and scalability of our original data collection protocol. We agree that the initial design—which included 9 auscultation points, 6 distinct body postures, and 30-second recordings per site—resulted in a substantial dataset (~54 minutes per participant) and would pose challenges for real-world implementation, particularly in resource-limited or primary care settings.
However, as this was the inaugural study of host-response vibroacoustics, our primary goal was to learn, acquire a rich, high-dimensional dataset that would allow us to interrogate the influence of posture-related hydrostatic pressure differentials and their effects on chest wall resonance and biofield signal propagation. This comprehensive sampling approach enabled us to explore positional variation in signal quality and identify those configurations that most reliably capture diagnostic biosignatures.
Based on the insights gained from this foundational dataset, we were able to rationally down-select to a simplified and clinically practical 6-step anterior chest protocol. This refined protocol maintains high diagnostic signal fidelity while dramatically reducing acquisition time, positional complexity, and data volume—thereby enhancing scalability and ease of use in point-of-care and lower-resource environments.
We have added a statement to the manuscript (p15, line 394) explicitly acknowledging the limitations of the initial protocol and clarifying that future clinical implementations will use this optimized, simplified version. This progression from exhaustive exploration to focused application is consistent with the translational pathway of many emerging physiological sensing modalities.
The study only assessed short-term signal variations (with two visits 24–72 hours apart) and lacked long-term follow-up, leaving the long-term validity and stability of these biomarkers unknown for disease progression, treatment monitoring, or recovery assessment. For example, whether the signals change predictably with clinical improvement or deterioration, or whether the method can dynamically track disease course, remains unaddressed—critical gaps for clinical applicability. The authors should acknowledge this limitation in the manuscript.
Response: Agreed and added to manuscript (p15, line 409). *Of note in our longitudinal TB studies we have now shown the trajectory of host-response vibroacoustics as a measure of disease severity time course.
The study population comprised only hospitalized symptomatic patients, whereas one of the stated goals was early or pre-symptomatic detection. The absence of asymptomatic or early-stage pre-symptomatic cases limits validation of the method’s effectiveness for true early detection, restricting its potential for disease prevention and control. The authors should acknowledge this limitation in the manuscript.
Response: We agree and have added this limitation in the manuscript (p15, line 388).
With only 19 COVID-19 hospitalized patients and 16 matched controls, the small sample size may compromise statistical power and fail to represent real-world diversity (e.g., across age, sex, comorbidities, disease severity), limiting generalizability. The authors should explicitly state this limitation.
Response: We have highlighted this limitation in the manuscript (p15, line 394).